# A Sensitive and Transparent Method for Tumor-Informed Detection of Circulating Tumor DNA in Ovarian Cancer Using Whole-Genome Sequencing

**DOI:** 10.3390/ijms252413349

**Published:** 2024-12-12

**Authors:** Christine Fribert Thusgaard, Sepideh Sadegh, Kirsten Marie Jochumsen, Torben Arvid Kruse, Mads Thomassen

**Affiliations:** 1Department of Gynecology and Obstetrics, Odense University Hospital, 5000 Odense, Denmark; kirsten.jochumsen@rsyd.dk; 2Research Unit of Gynecology and Obstetrics, Department of Clinical Research, University of Southern Denmark, Odense University Hospital, 5000 Odense, Denmark; 3Department of Clinical Genetics, Odense University Hospital, 5000 Odense, Denmark; sepideh.sadegh@rsyd.dk (S.S.); torben.kruse@rsyd.dk (T.A.K.); mads.thomassen@rsyd.dk (M.T.); 4Clinical Genome Center, Department of Clinical Research, University of Southern Denmark, 5000 Odense, Denmark

**Keywords:** circulating tumor DNA, epithelial ovarian carcinoma, carcinoma, whole-genome sequencing, liquid biopsy, single-nucleotide variant, single-nucleotide variant detection

## Abstract

Circulating tumor DNA (ctDNA) is a biomarker that could potentially improve the survival rate of ovarian cancer (OC), e.g., by monitoring treatment response and early relapse detection. However, an optimal method for ctDNA analysis in OC remains to be established. We developed a method for tumor-informed single-nucleotide variant detection of ctDNA in OC using whole-genome sequencing. Tumor and plasma samples obtained at the time of diagnosis from 10 patients with OC were included. The tested method involved applying basic filters with different cut-offs of read depth, allelic depth, and variant allele frequency of tumor and normal DNA. In addition, we applied a new filtering approach using plasma samples from the other included OC patients (the plasma pool) for specific removal of artefacts. The basic filters with varying cut-offs showed minor improvement in signal-to-noise ratio (S2N). However, the addition of the plasma pool filter resulted in a considerable ctDNA signal improvement, indicated by both S2N and z-score. This study demonstrates a promising method for ctDNA detection in OC patients using a tumor-informed approach for whole-genome sequencing. Despite the limited number of patients involved, the results suggest a significant potential of the method for ctDNA signal detection in patients with OC.

## 1. Introduction

Circulating tumor DNA (ctDNA) has been identified as a promising biomarker in certain types of cancer, including ovarian cancer (OC). CtDNA is the fraction of cell-free DNA (cfDNA) in plasma released from a malignant tumor into the circulation [1,2]. A low fraction of ctDNA in cfDNA is a limitation in ctDNA detection. Previously, the consensus has been to increase the sequencing depth of a limited number of targets to improve the detection of ctDNA. However, ctDNA is not always detected by ultra-deep targeted sequencing, especially in cases with a low tumor burden [3]. Recent studies have shown that the breadth of sequencing complements the sequencing depth, and the widest sequencing breadth is achieved through whole-genome sequencing (WGS) [3,4,5,6,7]. By using WGS to detect single-nucleotide variants (SNVs) in ctDNA, a study detected ctDNA with tumor fractions as low as 10^−5^, compared to 10^−3^ with the use of ultra-deep targeted sequencing [3]. WGS has also been used to detect ctDNA in patients with OC, but the number of studies is limited, and the methods vary among the studies [8,9,10,11]. Furthermore, the use of WGS for ctDNA detection focusing on SNVs in OC has not yet been reported. The limited tools available for ctDNA detection through WGS are often complex and challenging to use.

OC accounted for 3.4% of cancer cases and 4.7% of cancer-related deaths in women in 2020 [12]. High-grade serous carcinoma of the ovaries, fallopian tubes, and peritoneum is the most lethal gynecological cancer. It is the most common type of OC [13,14] and, because of the aggressive nature of the disease, around two-thirds of the patients are diagnosed in advanced stages (FIGO stage III–IV) [14]. The treatment of ovarian cancer is a combination of surgery and chemotherapy, as well as targeted treatment with PARP inhibitors or bevacizumab for specific groups of patients, e.g., patients with somatic and germline *BRCA* mutations or homologous recombination deficiency [14,15]. Despite complete resection of advanced disease and treatment with chemotherapy, the majority of these patients will relapse within a few years [14], and the five-year survival rate is limited to 30–40% [16]. The current biomarker for epithelial OC, CA-125, lacks specificity and sensitivity [14], and the low survival rate indicates that a more specific and sensitive biomarker for OC could potentially increase the survival rate of the disease.

In this study, we established a sensitive and transparent method for tumor-informed SNV detection of ctDNA in the blood of OC patients using WGS, which to the best of our knowledge has not been done before. A tumor-informed method, using a combination of various allelic and read depth filters with different cut-offs, an additional panel of normals (leucocyte DNA), and a new filtering approach based on forced calling of tumor variants in a pool of plasma samples, was assessed to improve ctDNA signal detection in plasma. Subsequently, the utility of this method for ctDNA signal detection was tested at a reduced sequencing depth in plasma.

## 2. Results

An overview of the WGS data analysis pipeline and the three methods tested in this study is illustrated in Figure 1. A detailed description is included in the Materials and Methods (Section 4).

### 2.1. Clinical Data 

Samples from 10 patients with high-grade serous OC were included in this study. Because of the short half-life of ctDNA (less than two hours [17]), blood samples were collected before treatment was initiated. After this, all patients received primary surgery and adjuvant chemotherapy. The tumor samples were collected after surgical removal. Due to contamination of one of the tumor samples, one patient was excluded. Plasma samples from nine patients with benign ovarian tumors were used as control samples. Table 1 summarizes the clinical characteristics of the included patients and controls.

### 2.2. Basic Noise Filtering in Tumor-Informed Plasma Analysis

To establish the best filters for identification of mutations in plasma, signal-to-noise ratios (S2N) were computed. The S2N is the average variant allele frequency (VAF) or alternate allelic depth (AD) in the plasma for mutations identified in the tumor, divided by the average VAF (or AD) of the same mutations in the control plasma. The number of variants detected in the tumor and the S2N for all of the applied filter thresholds are presented in Appendix A. Increasing the filter threshold for the read depth (DP) of normal DNA and the DP, AD, and VAF of tumor DNA showed a limited improvement in VAF and AD signal-to-noise ratios (Figure 2, Appendix A). For the analyses presented in the following paragraphs, we applied the most optimal setting of filter thresholds (Figure 2, Filter E).

### 2.3. Advanced Filtering with Different Approaches and ctDNA Signal Detection

For the first part of this study, a panel of normals (PON) (1000_g PON) provided by GATK was used (Method 1). To test whether an additional PON would reduce the level of noise, we introduced a self-generated PON based on normal DNA samples from the patients included in this study. Using a merged PON (Method 2) increased the VAF S2N for all patients (Figure 3). The improvement ranged from a factor of 1.4 to 3.9 (Appendix A). Similar results were seen for AD S2N (Appendix A). We also calculated the z-score, a good statistical measure used to evaluate the strength of the signal relative to the noise. This is calculated based on the mean and standard deviation of the noise model (control plasmas). Further details can be found in Section 4.6. The z-scores for both VAF and AD increased after applying the merged PON (Figure 4 and Appendix A).

When inspecting the data from Methods 1 and 2 in detail, we expected VAF values close to zero in the control plasmas. However, in a relatively low number of positions we observed outliers noticeably above zero, indicative of general sequencing errors. To remove these errors and thereby decrease the level of noise even further, we introduced a plasma pool filter that removed variants if they were seen with AD > 1 in at least one plasma sample from the plasma pool of the other nine OC patients (Method 3). This further improved the S2N and z-score for both VAF (Figure 3 and Figure 4) and AD (Appendix A) in seven patients (Appendix A). The binary z-scores improved in four patients (Appendix A). Interestingly, the addition of the plasma pool filter increased the S2N and z-scores to the same level when only applying the 1000_g PON (Appendix A).

UpSet plots [18] were generated to provide an overview of the number of variants removed by each and more than one of the different filters. The plot for patient 1 is illustrated in Figure 5 (the plots for the rest of the patients are in Appendix A). In patient 1, the plasma pool filter removed 411 variants, most of which (370) were also removed by other filters such as AD and VAF in the tumor. However, by filtering out the few variants (41) that were only removed by the plasma pool filter, the S2N and z-score increased considerably. This pattern was also seen in other patients, and the improvement is noticeable from Method 2 to Method 3 in Figure 3, except for patients 3 and 6 (FIGO stage I), who showed no clear improvement.

### 2.4. Downsampling of Plasma

To test whether the ctDNA signal is detectable in plasma sequenced with a lower depth, we performed the SNV calling analysis when the plasma was downsampled to 40 percent. This was equivalent to a sequencing depth of around 20×. This reduced the VAF S2N and z-scores in all patients (Figure 6 and Appendix A). However, these values were still high, indicating identifiable ctDNA signals at a lower sequencing depth in the seven non-stage-I patients (Appendix A).

## 3. Discussion

This method study demonstrates a tumor-informed approach for SNV detection in ctDNA from the blood of OC patients using WGS. With WGS, we were able to use the combined pattern of all identified SNVs in the plasma to detect ctDNA signal. By focusing on how to mitigate noise through different strategies, we found that the plasma pool filter clearly improved ctDNA signal detection. The ctDNA signal was still detectable even when the sequencing depth of plasma was reduced to around 20×.

No somatic variant caller for tumor DNA is perfect. Some variants are missed, and false positive positions (noise) are detected as true variants. We need a method that is useful even when the ctDNA content is low. It is therefore especially important to reduce noise in order to distinguish true variants from noise. The first step of this study was to evaluate the use of basic noise filtering and its ability to reduce noise in plasma. By increasing the filter thresholds in normal and tumor DNA, the number of identified variants dropped dramatically, while only a slight improvement was seen in S2N. The levels were therefore set with the balance between reducing noise and removal of the fewest true tumor variants in mind. These thresholds are not necessarily fixed and can depend on cancer type, tumor content, and the applied sequencing depth.

By applying the merged PON (Method 2), we observed a clear improvement. This indicates the effectiveness of applying a self-generated PON with normal samples that have been sequenced similarly to the tumor and plasma samples to reduce the level of noise. Our self-generated PON included ten normal DNA samples and was able to make a significant improvement. Increasing the number of these samples might improve noise filtering even further.

In patients with localized primary or recurrent cancer, it is likely that the ctDNA level is low; hence, a useful method for ctDNA detection should be sensitive enough to identify these cases. To obtain sensitive ctDNA analysis, it is vital that the applied method is able to remove background noise. To reduce the level of noise even further, we introduced a plasma pool filter consisting of plasma samples from the other OC patients included in the study, and we removed variants seen in at least one of the plasma pool samples (with AD > 1) from the patient plasma and from the control plasmas before further computation (Method 3). The plasma pool filter focuses on the variants identified in the tumor and assesses the level of noise in each variant position by performing forced variant calling of the tumor variants in all of the plasma pool samples. This allows for more sensitive noise reduction compared to other methods that use a general list of artefactual variants for noise removal, such as the plasma blacklist filter described as part of MRD-EDGE [7]. Even though the used plasma pool consisted of only nine patients, this filter showed the greatest improvement in ctDNA signal in seven out of nine patients, while removing relatively few variants. This was independent of whether the filter was applied before or after the addition of the merged PON (Method 2). When applying the plasma pool filter, it is important that the control plasma samples are treated in an identical way. All plasma samples were sequenced in the same way, and the plasma samples from the OC patients even in the same sequencing run. The plasma samples were therefore comparable, and there was only a slight variation in sequencing depth among them. The average sequencing depth was 52× among the plasmas from OC patients and 49× for the control plasmas. Furthermore, the computation and the assessment of variants identified in the patient’s own plasma, the plasma pool, and the control plasmas were run in identical manners. There is a slight risk that a true recurrent mutation could be removed by the plasma pool filter, but as recurrent mutations are rare, this would only lead to a minimal reduction in the signal.

In the current study, the z-score was calculated based on VAF, AD, and in a binary version. The binary z-score has been used in some other studies on ctDNA [3,7]. Our results showed that the introduction of plasma pool filtering in Method 3 improved the S2N, VAF, and AD z-scores. In contrast, the binary z-score showed no clear improvement (Appendix A). This addition of the plasma pool filter removed noisy outliers; thus, the results with the least amount of noise were extracted after the application of Method 3.

The sensitivity of the method was mainly determined by the average VAF in the noise model (Appendix A). After application of the plasma pool filter (Method 3), the level of noise was calculated as average VAF with values 3–4 times 10^−4^ in all patients, except patient 9. This was independent of the set of variants called, giving an indication of the maximum obtainable sensitivity. Seven of the samples showed signals far above the noise level. In the last two samples, the signals were two- and three-fold the level of noise, which was still significant at the 5% level (z-score higher than 1.895, one-sided). For patient 6, it was even significant at the 1% level.

The limitations of this study are the small number of patients included and the fact that all tumor samples had a very favorable tumor cell fraction of at least 75%. The method therefore needs to be validated on a higher number of patients with different tumor cell fractions. Furthermore, the method should be tested for different clinical applications. A tumor-informed approach such as the method applied in this study could potentially be useful for monitoring of treatment response, and in early detection of recurrence.

In conclusion, this method study is the first to demonstrate a successful method for ctDNA detection in OC patients using a tumor-informed approach for WGS focusing on SNVs. Despite the relatively low number of patients involved in this study, the results suggest a significant potential of our method in noise removal and ctDNA signal detection. It is therefore promising in the search for a biomarker in OC that might improve the survival rate of the disease. Furthermore, this method is readily comprehensible and could easily be applied and tested on other cancer types.

## 4. Materials and Methods

### 4.1. Inclusion of Patients

Patients with suspected primary OC were enrolled in the study from January 2016 to May 2017 at the Department of Gynecology and Obstetrics, Odense University Hospital, after giving written informed consent. From each patient, a pretreatment venous blood sample was collected in two Cell-Free DNA BCT tubes (STRECK, 230470, La Vista, NE, USA), and a frozen tissue sample was obtained from the tumor. Among the enrolled patients, 10 patients with high-grade serous OC and a tissue sample with a tumor cell fraction of at least 75%, as assessed by the pathologist, were selected for further analysis.

### 4.2. Sample Processing

Plasma and buffy coat (normal peripheral blood cells) were isolated from the blood samples and frozen within three days of collection. CfDNA was extracted from approximately 4 mL of plasma using the QIAamp Circulating Nucleic Acid Kit (QIAGEN, 55114, Venlo, The Netherlands), normal DNA was isolated using the Maxwell RSC Blood DNA (Promega, AS1400, Madison, WI, USA), and tumor DNA was extracted using the Maxwell RSC Tissue DNA Kit (Promega, AS1610, Madison, WI, USA), according to the manufacturers’ instructions. The Qubit dsDNA HS assay kit (Thermo Fisher Scientific, Q32854, Waltham, MA, USA) was used to assess the concentration of isolated DNA. To prepare samples for NGS, the ThruPLEX TagSeq HV Kit (Takara Bio, R400743, Kusatsu, Japan) was used on extracted cfDNA, tumor DNA, and normal leucocyte DNA, following the manufacturer’s instructions. Unique dual indices (ThruPLEX HV UDI Set A (Takara Bio, 400738, Kusatsu, Japan) were used to enable the removal of hopped reads from the downstream analysis.

### 4.3. Pre-Processing of WGS Data

Samples were sequenced on an Illumina NovaSeq 6000 with paired-end 2 × 150 base pairs. The yielded average sequencing coverage was 17× for normal DNA, 24× for tumor DNA, 52× for patient plasma, and 49× for control plasma. An overview of the computational analysis pipeline is illustrated in Figure 1. The raw sequenced data were pre-processed by first demultiplexing using the bcl2fastq tool (version 2.19.0) [19] and adapter trimming with Skewer software (version 0.2.2) [20] before the alignment to the human reference genome (GRCh38) using the BWA-MEM tool (version 0.7.17) [21]. Next, SortSam from Picard (version 4.2.6.1) [22] was used to sort and index the aligned files. This was followed by removing duplicates from the BAM files using MarkDuplicates from the GATK package (version 4.2.6.1) [23]. In addition to performing the quality control of the samples’ data with standard tools such as FastQC [24], Qualimap [25], and MultiQC [26], the NGSCheckmate tool (version 1.0.1) [27] was used to validate sample identity. This indicated misidentification or contamination of patient 4′s tumor sample. Therefore, this sample was eliminated from the study. However, the plasma from this patient passed the identity check and was included in the plasma pool filter (see Section 4.5). To simulate a reduced sequencing depth (around 20×) in plasma, a downsampling tool (DownsampleSam from PICARD (version 3.0.0) [28]) was used on the BAM files to randomly include 40 percent of the reads from the original WGS data (~50×).

### 4.4. Somatic Variant Calling

Mutect2 from the GATK package (version 4.2.6.1) [29] was used for SNV calling. First, Mutect2 was run on the tumor and matched normal BAM files to call somatic variants while using the PON provided by GATK (1000_g PON) for pre-calling filtering. Using the FilterMutectCalls method from GATK, we filtered out variants that were due to contamination (using the contamination table generated by GATK’s CalculateContamination function) and sequence context artefacts (generated by GATK’s LearnReadOrientationModel function). To implement the tumor-informed strategy for variant calling in plasma, Mutect2 was executed utilizing the variants identified in the corresponding tumor as the input set of alleles for force calling. To generate the noise model per patient, the variants detected in the patient’s tumor were force-called on every control plasma sample.

### 4.5. Different Methods for Filtering Variants

The vcf files from Mutect2 were imported to VarSeq (version 2.4.0) (Golden Helix Inc., Bozeman, MT, USA) [30]. After applying a gnomAD filter (gnomAD Genomes and Exomes variant frequencies 2.0.1) with AD ≤ 0.00001 in VarSeq, a concatenated table of all vcf files, including the information on the read depth (DP), AD, and VAF for their own tumor, normal, plasma, control plasmas, and the plasma pool (plasma from other OC-patients), was exported per patient. To assess the effects of basic filters, varying filter thresholds on DP, AD, and VAF for the tumor and normal DNA were applied to the Mutect2 results using Python scripts developed in-house. For this, the 1000_g PON was used in the variant calling (Method 1).

For more effective capturing and removal of technical artefacts, an additional PON from the normal DNA of the 10 OC patients, prepared by the same library kit and sequenced by the same technology as used for tumors and plasma, was generated. This self-generated PON was merged with the PON provided by Mutect2 and used in the first step of the variant calling pipeline (Method 2). To reduce the noise level further, we introduced the plasma pool filter that removed variants detected in at least one plasma sample from the nine other OC patients with AD > 1 (Method 3) (Figure 1).

### 4.6. ctDNA Signal Detection

To assess the ctDNA signal in plasma, we calculated the S2N and z-scores. S2N is calculated as the average of VAF (or AD) in the plasma for mutations identified in the tumor (after applying filters), divided by the average VAF (or AD) of the same mutations in the control plasmas (constituting our noise model):(1)$$S2N=\frac{Ave_{VarDet}}{\mu_{noise}}$$
where $Ave_{VarDet}$ is the average VAF (or AD) of detected variants in the patient’s plasma, and $\mu_{noise}$ is the mean VAF (or AD) in controls. Similarly, z-scores were calculated based on the mean and standard deviation of the noise model as follows:(2)$$z-score=\frac{Ave_{VarDet}-\mu_{noise}}{\sigma_{noise}}$$
where $\sigma_{noise}$ is the standard deviation of VAF (or AD) in controls. We also computed a binary z-score, i.e., the values of VAF and AD of the variants were not included in the calculation, only the number of detected variants.

## Figures and Tables

**Figure 1 ijms-25-13349-f001:**
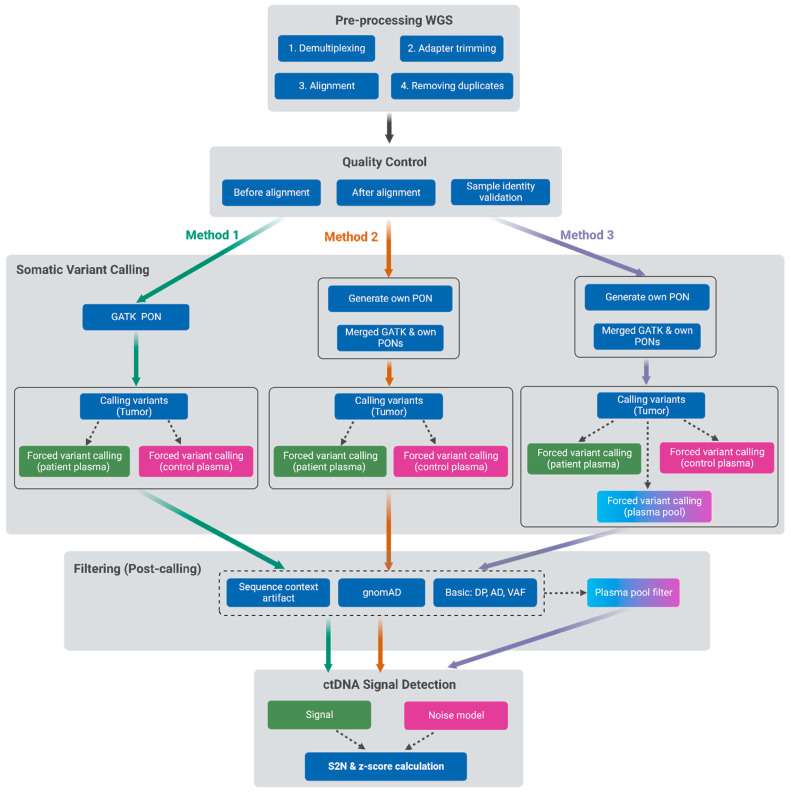
Overview of the Whole-Genome Sequencing (WGS) data analysis pipeline for ctDNA detection. Method 1: 1000_g panel of normals (PON); Method 2: merged PON; Method 3: merged PON and plasma pool filter. Read Depth (DP), Allelic Depth (AD), variant allele frequency (VAF).

**Figure 2 ijms-25-13349-f002:**
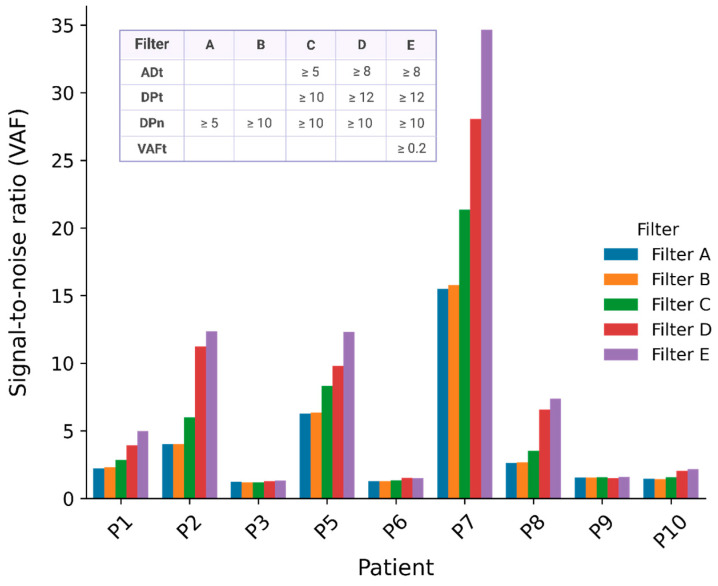
Overview of VAF signal-to-noise ratios after applying different filters: The stringency of the filter parameters increases progressively from Filter A to E as the filter threshold is altered for AD and DP of tumor and normal DNA, as well as for VAF of tumor DNA; 1000_g PON (Method 1) and ADn = 0 are applied to all. ADt (allelic depth in tumor), DPt (read depth in tumor), VAFt (variant allele frequency in tumor), ADn (allelic depth in normal), DPn (read depth in normal).

**Figure 3 ijms-25-13349-f003:**
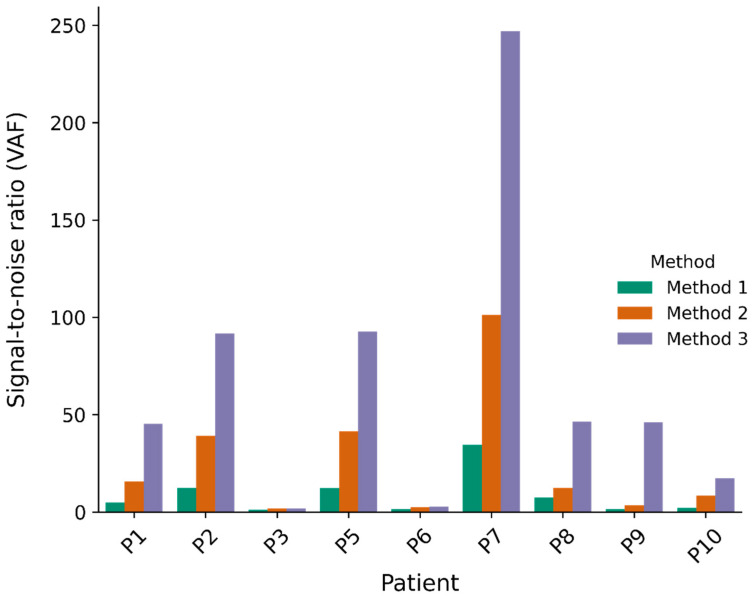
Overview of VAF signal-to-noise ratios when different methods for filtering noise are applied. Applied filtering methods: Method 1: 1000_g PON; Method 2: merged PON; Method 3: merged PON and plasma pool filter. Filtering thresholds ADt ≥ 8, DPt ≥ 12, Adn = 0, DPn ≥ 10, and VAFt ≥ 0.2 were used in all three methods.

**Figure 4 ijms-25-13349-f004:**
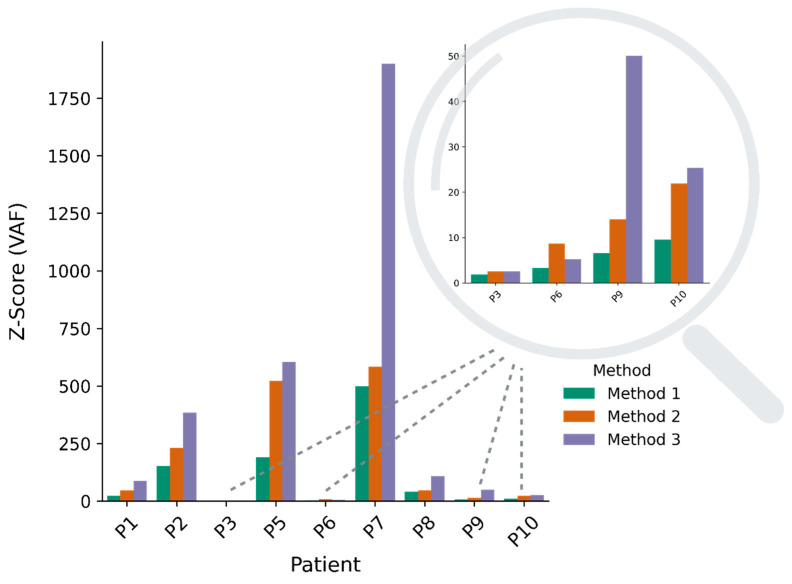
Overview of z-scores computed based on VAF when different methods for filtering noise are applied. The magnifying glass enlarges the z-score (VAF) in the four patients (P3, P6, P9, and P10) with low z-score results compared to the other five patients. Applied filtering parameters: Method 1: 1000_g PON; Method 2: merged PON; Method 3: merged PON and plasma pool filter. Filtering thresholds ADt ≥ 8, DPt ≥ 12, Adn = 0, DPn ≥ 10, and VAFt ≥ 0.2 were used in all three methods.

**Figure 5 ijms-25-13349-f005:**
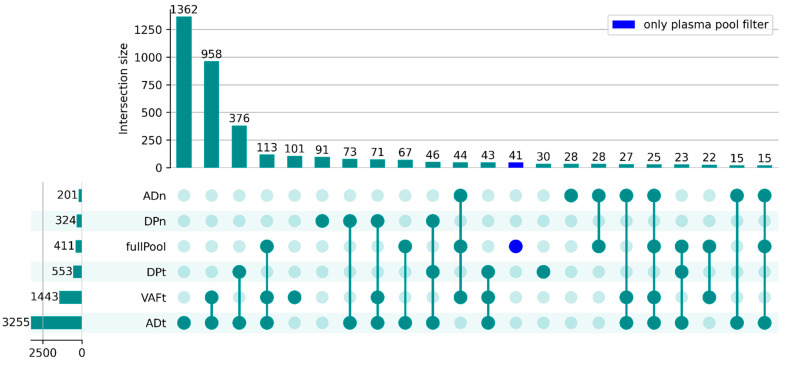
Overview of the number of variants removed by filters and the overlap between them for patient 1: UpSet plot shows the number of variants filtered by each filter, as well as the intersection between different filters. Intersection combinations with fewer than 15 variants are not shown. Applied filter thresholds: ADt ≥ 8, DPt ≥ 12, ADn = 0, DPn ≥ 10, VAFt ≥ 0.2, and fullPool (plasma pool). The merged PON was applied.

**Figure 6 ijms-25-13349-f006:**
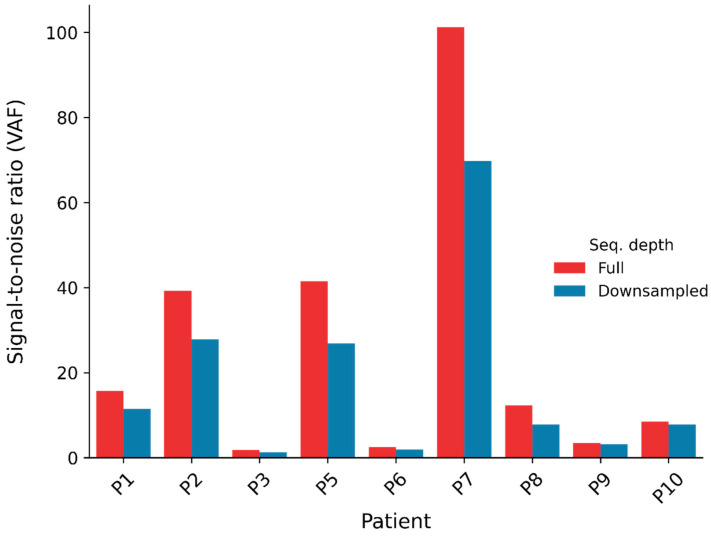
Overview of VAF signal-to-noise ratios for sequenced data at full and reduced depth (plasma downsampled to approximately 20×). The merged PON (Method 2) was applied. The plasma pool filter was not used.

**Table 1 ijms-25-13349-t001:** Clinical characteristics of patients and controls.

Patient Data	Number of Patients (9)	Number of Controls (9)
Pathology		
HGSC	9	
Serous adenoma		4
Mucinous adenoma		1
Fibroma		1
Other		3
Age at diagnosis (years)		
<65	6	6
≥65	3	3
FIGO stage		
I	2	NA
II	1	NA
III	3	NA
IV	3	NA

High-grade serous carcinoma (HGSC); not applicable (NA); FIGO stage I: tumor confined to ovaries; FIGO stage II: tumor involves one or both ovaries with pelvic extension (below the pelvic brim) or primary peritoneal cancer; FIGO stage III: tumor involves one or both ovaries with cytologically or histologically confirmed spread to the peritoneum outside the pelvis and/or metastasis to the retroperitoneal lymph nodes; FIGO stage IV: distant metastasis excluding peritoneal metastasis [14].

## Data Availability

Our ethical approval does not allow for the sharing of supporting sequencing data. The scripts used for bioinformatics analysis are available upon reasonable request.

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
