# Peer review of "A Sensitive and Transparent Method for Tumor-Informed Detection of Circulating Tumor DNA in Ovarian Cancer Using Whole-Genome Sequencing"

_ijms, 2024, doi:10.3390/ijms252413349_

Round 1
Reviewer 1 Report
Comments and Suggestions for Authors
The main focus of the manuscript entitled 'A sensitive and transparent method for tumor-informed detection of circulating tumor DNA in Ovarian Cancer using whole genome sequencing' is the development of a method for tumor-informed single nucleotide variant detection of ctDNA in Ovarian Cancer using whole-genome sequencing. This novel method, if tested on a larger number of patients, can be of significant help and utility. All sections are well-explained and the figures are high quality with thorough explanations. However, there are some concerns that I will address below:
Figures 3, 4, and 5: The authors mention a threshold, but do not provide an explanation on how it was calculated. Please provide a short explanation of the calculation method, and clarify the basis for determining the threshold value.
Line 243: In the '4.3. Pre-processing of WGS data' section, please add references for all software used.

Reviewer 2 Report
Comments and Suggestions for Authors
Dear Authors,
The manuscript presents a unique and robust approach for Single Nucleotide Variant (SNV) detection in circulating tumor DNA (ctDNA) of ovarian cancer patients using whole genome sequencing (WGS). The described filtering method, which leverages various sequencing parameters and control plasma specimens, appears to significantly enhance the specificity and sensitivity of variant calling. However, the limited number of samples analyzed in this study restricts the ability to draw definitive conclusions. The inclusion of additional prospective samples as a validation cohort would greatly strengthen the findings and their potential clinical relevance.
Below are additional points for clarification and improvement:
1-Clinical Relevance of Variants:
After applying the filtering steps, were the retained variants evaluated for their clinical significance in ovarian cancer? Did the authors identify clinically actionable mutations, such as driver or resistance mutations, in the ctDNA? Additionally, were any potentially significant variants removed as a result of the more stringent filtering criteria? Including such details would highlight the practical utility of the method.
2-Clinical Application:
A discussion on the potential clinical applications of this method would benefit readers. For example, how might this approach be used for disease monitoring or relapse detection in a clinical setting? Can this method be applied to identify driver variants in ctDNA without matched tumor samples, or to develop an SNV-based biomarker or signature for ovarian cancer?
3-Genomic SNV Signature:
Did the analysis identify any common SNV signatures across the tested samples? If so, discussing such findings could enhance the manuscript's contribution to understanding ctDNA profiles in ovarian cancer.
Overall, this manuscript demonstrates strong potential, and addressing the above points would significantly enhance its impact and clarity.
Sincerely,
Reviewer 3 Report
Comments and Suggestions for Authors
In the current manuscript titled "A sensitive and transparent method for tumor-informed detection of circulating tumor DNA in Ovarian Cancer using whole-genome sequencing" by Christine F. Thusgaard et al, the authors have elucidated using circulating tumor DNA (ctDNA) as a biomarker to access in ovarian cancer (OC) patients and improve their survival rates. In this study, using three methods and tumor samples from 10 patients as well as the plasma pool samples, the authors have demonstrated a promising method for ctDNA detection in OC patients using a tumor-informed approach for whole genome sequencing. While this manuscript offers a novel method for ctDNA detection and its use as a biomarker for signal detection in OC patients, there are following comments that need to be addressed to further improve the quality of this manuscript:
1. Introduction: What are the statistics of OC? What are the currently available treatments for OC? What are the effects, benefits and toxicities from these treatments? Are there any clinical trials ongoing with targeted therapies for OC? What is the survival rate of the patients with such treatments? Please describe briefly.
2. Table 1: Were these patients on prior treatments or underwent surgeries? Did any patients have metastasis reported? Describe in the table or in the results section or consider including this information in the material and methods section.
3. Figure 4: Describe the zoomed in image (shown in the figure) in the figure legend.
4. Material and methods 4.1: Consider including the clinical study protocol number.
5. Provide the catalog numbers and company names of the reagents and kits used in the material and methods section.
